# Prediction of Strong Transversal s(TE) Exciton–Polaritons in C_60_ Thin Crystalline Films

**DOI:** 10.3390/ijms23136943

**Published:** 2022-06-22

**Authors:** Vito Despoja, Leonardo Marušić

**Affiliations:** 1Institut za Fiziku, Bijenička 46, 10000 Zagreb, Croatia; 2Donostia International Physics Center (DIPC), P. Manuel de Lardizabal, 4, 20018 San Sebastián, Spain; 3Maritime Department, University of Zadar, M. Pavlinovića 1, 23000 Zadar, Croatia; lmarusic@unizd.hr

**Keywords:** eksciton-polaritons, molecular crystals, optical conductivity, photonics

## Abstract

If an exciton and a photon can change each other’s properties, indicating that the regime of their strong bond is achieved, it usually happens in standard microcavity devices, where the large overlap between the ’confined’ cavity photons and the 2D excitons enable the hybridization and the band gap opening in the parabolic photonic branch (as clear evidence of the strong exciton–photon coupling). Here, we show that the strong light–matter coupling can occur beyond the microcavity device setup, i.e., between the ’free’ s(TE) photons and excitons. The s(TE) exciton–polariton is a polarization mode, which (contrary to the p(TM) mode) appears only as a coexistence of a photon and an exciton, i.e., it vanishes in the non-retarded limit (c→∞). We show that a thin fullerene C${}_{60}$ crystalline film (consisting of *N* C${}_{60}$ single layers) deposited on an Al${}_{2}$O${}_{3}$ dielectric surface supports strong evanescent s(TE)-polarized exciton–polariton. The calculated Rabi splitting is more than Ω=500 meV for N=10, with a tendency to increase with *N*, indicating a very strong photonic character of the exciton–polariton.

## 1. Introduction

The interaction between photons and polarization modes can result in the formation of hybrid photon polarization modes, called polaritons [1]. Very common platforms for studying strong light–matter interactions are the gapped systems, such as semiconductors [2,3] or molecules [4], placed in microcavity devices, where the cavity exciton–polaritons are formed. The quantum nature of a cavity exciton–polariton manifests in the form of the Bose–Einstein condensation, which has recently been experimentally detected [2,5,6]. The cavity exciton–polaritons are routinely observed in bulk [7,8,9,10] and quantum well systems [3,11], e.g., devised from GaAs [3]. Two-dimensional (2D) materials, such as semiconducting monolayers, thin heterostructures, and films, are even more attractive than their bulk counterparts, due to the reduced Coulomb screening and the corresponding large exciton binding energies [12,13,14,15,16,17,18,19] that enable the formation of well-defined exciton–polaritons even at room temperatures [20]. The first 2D exciton–polaritons were obtained in a monolayer of a transition metal dichalcogenide (TMD) MoS${}_{2}$, where the Rabi splitting between the exciton and the cavity photon of ∼50 meV was observed [21]. Further photoluminescence studies showed clear anti-crossing behavior and splitting of the exciton–polariton in other 2D TMD cavity devices, e.g., in MoSe${}_{2}$ [22], WS${}_{2}$ [23], WSe${}_{2}$ [24,25], and in the MoSe${}_{2}$-WSe${}_{2}$ heterostructure [26]. In addition, the real-space imaging of the exciton–polaritons has been done by means of near-field scanning optical microscopy for WSe${}_{2}$ thin films [27]. Finally, a remarkable Rabi splitting of 440 meV was theoretically predicted in the monolayer hexagonal boron nitride cavity device [28], suggesting extraordinary strong light–matter interaction in 2D heterostructures.

The strongest exciton–photon coupling is achieved in the organic dye molecule thin films placed in a microcavity [29]. For example, Rabi splitting of Ω≤450 meV [30,31,32], 0.7 eV [33], and even more than 1 eV [34,35] have been detected when various organic dye molecules were placed in a planar microcavity. The theoretical approach to such systems is mostly based on the two-level or boson–boson Hamiltonian model with an arbitrary coupling constant [4,36]. Using graphene [37] or perovskite [38]-layered heterostructures, one can obtain tunable microcavity devices of high performance, which can be applied as photonic detectors or emitters [38,39], but also as platforms for studying the exciton–photon coupling.

All these studies use the same concept: an exciton in a semiconducting nanostructure hybridizes with a photon ’confined’ in a metallic microcavity. Such cavity photons and excitons are expected to interact stronger as the photon is more confined (i.e., the overlap between the exciton and photon is larger) and as the exciton oscillator strength [28] is larger. In this paper, we change the concept and explore the coupling between the ’free’ photons and the excitons in the 2D nanostructures. The coupling between the ’free’ photons and the polarization modes (such as plasmons, phonons, or excitons at surfaces or in 2D nanostructures) is a well-known, widely explored phenomenon [40,41,42,43,44,45,46,47]. The inherent property of all these eigenmodes (called plasmon polaritons, phonon polaritons, or exciton–polaritons) is their evanescent character, i.e., they are well-defined eigenmodes with the electromagnetic field (wave function) strongly localized at the interface or within the 2D nanostructure. Moreover, these modes usually have p or transverse-magnetic (TM) polarization, i.e., the electric field (and, therefore, the currents as well) has a component parallel to the direction of propagation. The most relevant point is that the p(TM) polarized plasmon polariton, phonon polariton, or exciton–polariton branches reduce to the plasmon, phonon, or exciton branches in the non-retarded (c→∞) limit. On the other hand, the electric field and the current of s or transverse-electric (TE) electromagnetic eigenmodes are perpendicular to the direction of their propagation. An especially attractive aspect of these polarization modes is that they **do not** exist in the non-retarded (c→∞) limit, i.e., they appear only as the coexistence of a photon and an exciton. The extent of the photon’s participation in the s(TE) exciton–polariton is determined from the bending of the horizontal exciton branch ($\omega_{\mathrm{ex}}$) at the exciton–photon crossing ($\omega_{\mathrm{ex}}=Q_{\mathrm{ex}}c/\sqrt{\epsilon }$, where $Q_{\mathrm{ex}}$ is the photon wave vector at the exciton–photon crossing point), which we call the Rabi splitting Ω, to keep the terminology compatible with the cavity systems. Even though the s(TE) surface or 2D polaritons do exist [40] for some conditions, there is still no experimental evidence of such modes. However, the hybridization between the s(TE) Bloch surface waves (BSWs) (i.e., the photons confined between a truncated photonic crystal and a semi-infinite dielectric), and the excitons has been experimentally demonstrated in both inorganic (quantum well and TMD monolayer) and organic systems [48,49,50,51].

We show that very strong s(TE) exciton–polaritons may occur in layered van der Waals (vdW) heterostructures. The prototypical layered nanostructure we investigate in this paper is a thin film of the FCC fullerite (crystalline fullerene) cut along the (111) planes so that it formed several (*N*) molecular (C${}_{60}$) layers. The crystalline C${}_{60}$ films were also deposited on a dielectric Al${}_{2}$O${}_{3}$ surface to make the simulation more realistic. The epitaxial growth of the C${}_{60}$ thin films of various thicknesses on various metallic or dielectric substrates under ambient conditions and in high vacuum was studied in references [52,53,54,55,56,57,58]. Some experimental studies even show that the crystalline growth of the C${}_{60}$ thin films on pentacene buffer layers is exclusively (111)-oriented [55]. Theoretical, molecular dynamic simulations of the C${}_{60}$ multilayer epitaxial growth and stability on various substrates were investigated in references [58,59,60,61]. These experimental/theoretical studies suggest that our model system is indeed highly realistic.

In this paper, the light–matter interaction was studied using our quantum electrodynamic Bethe–Salpeter equation approach (QE-BSE) developed in references [28,47]. This approach describes both excitons and photons by bosonic propagators σ and Γ, respectively, derived from the first principles. The C${}_{60}$ optical conductivity σ was calculated using *ab initio* G${}_{0}$W${}_{0}$-BSE method [47,62], and the free proton propagator Γ was derived by solving Maxwell’s equation at the vacuum/dielectric interface [63,64]. The dielectric surface was described by the local dielectric function $\epsilon_{M}\left(\omega \right)$, also determined from the first principles. The exciton–photon coupling was achieved by dressing the free-photon propagator Γ with excitons at the random phase approximation (RPA) level. We studied the s(TE)-polarized exciton–polariton in the C${}_{60}$ thin crystalline film as a function of the number *N* of the C${}_{60}$ single layers. We obtained a very strong hybridization between the exciton in the C${}_{60}$ thin film and the s(TE) free photons. The hybridization increased with *N*, and for N=10, we achieved the Rabi splitting Ω even larger than 1000 meV and 500 meV for self-standing and supported C${}_{60}$ films, respectively.

The paper is organized as follows. In Section 2, we present the geometry of the system and the derivation of the optical conductivity $\tilde{\sigma }$ of the C${}_{60}$ single layer using the G${}_{0}$W${}_{0}$-BSE approach with the solution of Dyson’s equation for the electric field propagator $E=\Gamma +\Gamma \tilde{\sigma }E$. In Section 3, we present the spectra of the electromagnetic modes S=ReE, as well as the dispersion relations and intensities of the s(TE)-polarized exciton–polaritons in the C${}_{60}$ thin films of different thicknesses *N*, in vacuum, and at a Al${}_{2}$O${}_{3}$ surface. The conclusions are presented in Section 4.

## 2. Theoretical Formulation

We assume the the C${}_{60}$ molecules, upon deposition on the crystal surface, self-assemble in a regular FCC structure (the most stable bulk structure of crystalline fullerene) forming a (111) surface, as shown in Figure 1a. The FCC crystal lattice constant is taken to be $a_{3D}=14$ Å [61], and the separation between the layers is fixed to be $\Delta =a_{3D}/\sqrt{3}=8.1$ Å. Each crystal plane forms a 2D hexagonal Bravais lattice with the lattice constant $a_{2D}=a_{3D}/\sqrt{2}=9.9$ Å, as shown in Figure 1b. The $C_{60}$ films, occupying z>0 half-space, are immersed in a dielectric medium described by a dielectric constant $\epsilon_{0}$. The dielectric response of the substrate, occupying z<0 half-space, is approximated by a local macroscopic dielectric function $\epsilon_{M}\left(\omega \right)$.

### 2.1. Calculation of Electric Field Propagator E

The quantity we used to extract the information from the electromagnetic modes in the $C_{60}$ films deposited on a dielectric surface was the electric field propagator $E_{\mu \nu }$, which provides the electric field produced by an external oscillating point dipole $p_{0}e^{-i\omega t}$ placed at point $r^{\prime }$ as [63,64]
(1)$$E_{\mu }\left(r,\omega \right)=\sum_{\nu =x,y,z}E_{\mu \nu }\left(r,r^{\prime },\omega \right)p_{\nu }^{0}.$$The propagator E is the solution of Dyson’s equation [28,47,64,65]
(2)$$E_{\mu \nu }\left(r,r^{\prime },\omega \right)=\Gamma_{\mu \nu }\left(r,r^{\prime },\omega \right)+\sum_{\alpha ,\beta =x,y,z}\int dr_{1}\int dr_{2}\;\Gamma_{\mu \alpha }\left({r,r}_{1},\omega \right)\sigma_{\alpha \beta }\left(r_{1},r_{2},\omega \right)E_{\beta \nu }\left(r_{2},r^{\prime },\omega \right),$$
where the integration is performed over the entire space, σ is the nonlocal conductivity tensor of the deposited C${}_{60}$ thin film, and Γ is the electric field propagator in the absence of the C${}_{60}$ film, i.e., when σ=0 [64,65]. The propagator Γ also includes the electromagnetic field scattering at the medium/substrate interface. If each molecule is approximated as a point polarizable dipole, then the optical conductivity of the C${}_{60}$ film can be written as
(3)$$\sigma_{\mu \nu }\left(r,r^{\prime },\omega \right)=\sum_{i=1}^{N}\sum_{R_{\|}}\sigma_{i}^{\mu \nu }\left(\omega \right)\delta \left(\rho -R_{\|}\right)\delta \left(z-z_{i}\right)\delta \left(\rho^{\prime }-R_{\|}\right)\delta \left(z^{\prime }-z_{i}\right),$$
where $\sigma_{i}^{\mu \nu }\left(\omega \right)$ is the optical conductivity tensor of a single molecule in the *i*-th molecular layer. This approximation is fully justified in the optical limit $2\pi c/\omega_{\mathrm{light}}>>R_{M}$, where $R_{M}$ is the radius of a C${}_{60}$ molecule. Note that although all the molecules are equal, we distinguished between their conductivities in different layers $\sigma_{i};i=1,N$, due to the different influences of the substrates on a molecule in a different layer. The 2D Bravais lattice translation vectors spanning the molecular crystal are
(4)$$R_{\|}=na_{1}+ma_{2};\;\;n,m\in Z,$$
where $a_{1}$ and $a_{1}$ are the primitive vectors, as illustrated in Figure 1b. The molecular layers occupy the planes
$$z_{i}=z_{0},z_{0}+\Delta ,z_{0}+2\Delta ,\cdots ,z_{0}+\left(N-1\right)\Delta ,$$
where *N* is the number of molecular layers. Due to the planar translational invariance of the substrate, the Fourier transform of the propagator Γ is
(5)$$\Gamma_{\mu \nu }\left(r,r^{\prime },\omega \right)=\int \frac{dQ}{{\left(2\pi \right)}^{2}}\Gamma_{\mu \nu }\left(Q,\omega ,z,z^{\prime }\right)e^{iQ(\rho -\rho^{\prime })},$$
where $Q=(Q_{x},Q_{x})$ are the two-dimensional wave vectors. The propagator E should also include the effects of the electromagnetic field Bragg scattering at the 2D crystal lattice, so that its Fourier transform is
(6)$$E_{\mu \nu }\left(r,r^{\prime },\omega \right)=\sum_{g_{\|}}\int \frac{dQ}{{\left(2\pi \right)}^{2}}E_{g_{\|}}^{\mu \nu }\left(Q,\omega ,z,z^{\prime }\right)e^{iQ\rho }e^{-i\left(Q+g_{\|}\right)\rho^{\prime }},$$
where
(7)$$g_{\|}=nb_{1}+mb_{2};\;\;n,m\in Z$$
are 2D reciprocal vectors, while $b_{1}$ and $b_{1}$ are primitive reciprocal vectors. After inserting (Equation 3), (Equation 5), and (Equation 6) in (Equation 2), it becomes an equation in the (Q,ω,z) space
(8)$$\begin{array}{c} E_{g_{\|}}^{\mu \nu }\left(Q,\omega ,z,z^{\prime }\right)=\Gamma_{\mu \nu }\left(Q,\omega ,z,z^{\prime }\right)+\;\\ \sum_{\alpha ,\beta =x,y,z}\sum_{i}\sum_{g_{\|}^{\prime }}\Gamma_{\mu \alpha }\left(Q,\omega ,z,z_{i}\right){\tilde{\sigma }}_{i}^{\alpha \beta }\left(\omega \right)E_{g_{\|}+g_{\|}^{\prime }}^{\beta \nu }\left(Q-g_{\|}^{\prime },\omega ,z_{i},z^{\prime }\right),\; \end{array}$$
where the surface optical conductivity is
(9)$${\tilde{\sigma }}_{i}^{\alpha \beta }\left(\omega \right)=\frac{1}{S_{\mathrm{fcc}}}\sigma_{i}^{\alpha \beta }\left(\omega \right),$$
and $S_{\mathrm{fcc}}=\left(a_{1}\times a_{2}\right)\hat{z}=a_{2D}^{2}\sqrt{3}/2$ is the area of the 2D unit cell. If we neglect the electromagnetic field Bragg scattering, by introducing $g_{\|}=g_{\|}^{\prime }=0$ in Equation (Equation 8), and inserting $z=z_{i}$ and $z^{\prime }=z_{j}$, the equation becomes the matrix tensor equation: (10)$$E_{\mu \nu }\left(Q,\omega ,z_{i},z_{j}\right)=\Gamma_{\mu \nu }\left(Q,\omega ,z_{i},z_{j}\right)+\sum_{\alpha ,\beta =x,y,z}\sum_{k}\Gamma_{\mu \alpha }\left(Q,\omega ,z_{i},z_{k}\right){\tilde{\sigma }}_{k}^{\alpha \beta }\left(\omega \right)E_{\beta \nu }\left(Q,\omega ,z_{k},z_{j}\right),$$
where $E^{\mu \nu }\left(z_{i},z_{j}\right)$ is the electric field propagator within (i=j) or between (i≠j) the C${}_{60}$ layers. The electrical field propagator in the absence of the C${}_{60}$ film can be written as
(11)$$\Gamma =\Gamma^{0}+\Gamma^{sc},$$
where the propagator of the ’free’ electric field (or free photons propagator) is [63,64]
(12)$$\Gamma^{0}\left(Q,\omega ,z,z^{\prime }\right)=-\frac{4\pi i}{\epsilon_{0}\omega }\delta \left(z-z^{\prime }\right)z\cdot z-\frac{2\pi \omega }{\beta_{0}c^{2}}e^{i\beta_{0}\left|z-z^{\prime }\right|}\sum_{q=s,p}e_{q}^{0}\cdot e_{q}^{0}.$$The propagator of the scattered electric field in the region $z,z^{\prime }>0$ is [63]
(13)$$\Gamma^{\mathrm{sc}}\left(Q,\omega ,z,z^{\prime }\right)=-\frac{2\pi \omega }{\beta_{0}c^{2}}e^{i\beta_{0}\left(z+z^{\prime }\right)}\sum_{q=s,p}r_{q}\cdot e_{q}^{+}\cdot e_{q}^{-}.$$Here, the unit vectors of the s(TE)-polarized electromagnetic field are
$$e_{s}^{0,\pm }=Q_{0}\times z.$$
and the unit vectors of the p(TM) polarized electromagnetic field are
$$e_{p}^{0,\pm }=\frac{c}{\omega \sqrt{\epsilon_{0}}}\left[\alpha_{0,\pm }\beta_{0}Q_{0}+Qz\right],$$
where $\alpha_{0}=-sgn\left(z-z^{\prime }\right)$, $\alpha_{\pm }=\mp 1$, and $Q_{0}$ and z are the unit vectors in the Q and *z* directions, respectively. The reflection coefficients of the s(TE) and p(TM) polarized electromagnetic waves at the medium/substrate interface are
$$r_{s}=\frac{\beta_{0}-\beta_{M}}{\beta_{0}+\beta_{M}}$$
and
$$r_{p}=\frac{\beta_{0}\epsilon_{M}-\beta_{M}\epsilon_{0}}{\beta_{0}\epsilon_{M}+\beta_{M}\epsilon_{0}},$$
respectively, and the complex wave vectors in perpendicular (*z*) direction are
$$\beta_{0,M}=\sqrt{\frac{\omega^{2}}{c^{2}}\epsilon_{0,M}\left(\omega \right)-{\left|Q\right|}^{2}}.$$The $\beta_{0}$ and $\beta_{M}$ determine the character of the electromagnetic modes at the medium/dielectric substrate interface. To simplify the interpretation, we assume that the dielectric medium is vacuum, i.e., $\epsilon_{0}=1$, and that the dielectric function of the substrate is constant $\epsilon_{M}\left(\omega \right)=\epsilon_{M}$, which is a plausible approximation for many wide band gap insulators. In the vacuum, for ω>Qc, $\beta_{0}$ is a real number; therefore, the electromagnetic modes have a radiative character, and for ω<Qc, $\beta_{0}$ is an imaginary number so that the electromagnetic modes have evanescent character. The two regions are separated by the so-called ’light-line’ ω=Qc, as illustrated by the magenta line in Figure 2a. In analogy to that, the two regions in the substrate are separated by the $\omega =Qc/\sqrt{\epsilon_{M}}$ line, as illustrated by the green line in Figure 2a. Since $\epsilon_{M}>1$, the slope of the light-line in the substrate is smaller than in the vacuum, so in the gap $Qc/\sqrt{\epsilon_{M}}<\omega <Qc$, the light propagates freely into the substrate but has an evanescent character in the vacuum region, as illustrated in Figure 2a. Therefore, the exciton–polariton mode $\omega_{\mathrm{ex}-\mathrm{pol}}$ is expected to appear in the fully evanescent region $\omega <Qc/\sqrt{\epsilon_{M}}$, since in that region it cannot be irradiated into the surrounding media. The evanescent character of the electric field produced by the exciton–polariton in the C${}_{60}$ film is illustrated in Figure 2b.

We chose the electromagnetic modes to propagate in the $Q_{0}\;=y$ directions. In this case, the Dyson Equation (Equation 10) decouples into the independent matrix and the matrix tensor equations for the s(TE) and p(TM) polarizations, respectively. Here, we explore the s(TE)-polarized electromagnetic modes, which satisfy the matrix equation:(14)$$\begin{array}{c} E_{xx}(|Q|y,\omega ,z_{i},z_{j})=\Gamma_{xx}(|Q|y,\omega ,z_{i},z_{j})+\;\\ \sum_{k}\Gamma_{xx}(|Q|y,\omega ,z_{i},z_{k}){\tilde{\sigma }}_{k}^{xx}\left(\omega \right)E_{xx}(|Q|y,\omega ,z_{k},z_{j}),\; \end{array}$$
where after using (Equation 11)–(Equation 13)
(15)$$\Gamma_{xx}(|Q|y,\omega ,z_{i},z_{j})=-\frac{2\pi \omega }{\beta_{0}c^{2}}\left[e^{i\beta_{0}\left|z_{i}-z_{j}\right|}+r_{s}e^{i\beta_{0}\left(z_{i}+z_{j}\right)}\right].$$The first term in (Equation 15) represents the incident electromagnetic field, while the second term represents the one reflected at the vacuum/substrate boundary. In the electrostatic or non-retarded limit c→∞ and
(16)$$\begin{array}{c} \lim_{c\to \infty }\Gamma_{xx}(|Q|y,\omega ,z_{i},z_{j})=0, \end{array}$$
so that all the properties presented below are a direct consequence of the binding between the s(TE)-polarized light (photons) and the molecular excitons, and they vanish, i.e., do not exist in the electrostatic limit.

### 2.2. Calculation of the Optical Conductivity of a Single Molecule

First, we explain the calculation of the molecular conductivity $\sigma_{i}^{\mu \nu }\left(\omega \right)$ in a ‘self-standing molecule’($z_{0}\to \infty$), and then we extend that to the case when the molecule is close to a dielectric surface, i.e., when $z_{0}$ is finite, and chosen to be a characteristic vdW distance. The basic ingredients we need to calculate the molecular conductivity $\sigma_{\mu \nu }\left(\omega \right)$ are the molecular orbitals $\phi_{n}\left(r\right)$ and the energies $E_{n}$, which can be obtained by solving the Kohn–Sham equation self-consistently. We assume that the molecules are periodically repeating so that they form a simple cubic (sc) Bravais lattice with the unit cell *a* and volume $\Omega_{\mathrm{sc}}=a^{3}$. The unit cell *a* is chosen so that there is no intermolecular overlap. This allows the molecular states $|\phi_{n}\rangle$ to be calculated at the Γ point only. It should be emphasized that the sc lattice and the unit cell *a* are not related to the previously described FCC lattice with the unit cell ‘$a_{3D}$’. The purpose of the sc lattice is only to determine the molecular states $|\phi_{n}\rangle$ at the Γ point using the plane-wave DFT code. From now on, the conductivity of the 3D molecular crystal will be denoted as $\sigma_{\mu \nu }^{3D}\left(\omega \right)$.

The nonlocal optical conductivity tensor in the 3D molecular crystal is [62,64,65,66]
(17)$$\sigma_{\mu \nu }^{3D}\left(r,r^{\prime },\omega \right)=\frac{2i}{\omega }\sum_{nm}\sum_{n^{\prime }m^{\prime }}K_{n\to n^{\prime }}^{m\leftarrow m^{\prime }}\left(\omega \right)j_{nm}^{\mu }\left(r\right){\left[j_{n^{\prime }m^{\prime }}^{\nu }\left(r^{\prime }\right)\right]}^{*},$$
where
(18)$$j_{nm}^{\alpha }\left(r\right)=\frac{e\hbar }{2im}\left\{\phi_{n}^{*}\left(r\right)\partial_{\alpha }\phi_{m}\left(r\right)-\left[\partial_{\alpha }\phi_{n}^{*}\left(r\right)\right]\phi_{m}\left(r\right)\right\}$$
represents the current produced by the transition between the molecular states $|\phi_{n}\rangle \to |\phi_{m}\rangle$. Considering that the Bloch wave functions at the Γ point $\phi_{n}$ are periodic functions, tensor (Equation 17) can be expanded in the Fourier series
(19)$$\sigma_{\mu \nu }^{3D}\left(r,r^{\prime },\omega \right)=\frac{1}{\Omega_{\mathrm{sc}}}\sum_{{\mathrm{GG}}^{\prime }}e^{i\mathrm{Gr}}e^{-iG^{\prime }r^{\prime }}\sigma_{GG^{\prime }}^{\mu \nu }\left(\omega \right),$$
where the Fourier coefficients are
(20)$$\sigma_{GG^{\prime }}^{\mu \nu ,3D}\left(\omega \right)=\frac{i}{\omega }\frac{2}{\Omega_{\mathrm{sc}}}\sum_{nm}\sum_{n^{\prime }m^{\prime }}j_{nm}^{\mu }\left(G\right)K_{n\to n^{\prime }}^{m\leftarrow m^{\prime }}\left(\omega \right){\left[j_{n^{\prime }m^{\prime }}^{\nu }\left(G^{\prime }\right)\right]}^{*},$$
and the current vertices are
(21)$$j_{nm}^{\alpha }\left(G\right)\;=\;\int_{\Omega_{\mathrm{sc}}}dre^{-iGr}\;j_{nm}^{\alpha }\left(r\right).$$The four-point polarizability K can be obtained by solving the Bethe–Salpeter (BS) equation [28,47]
(22)$$K_{n\to n^{\prime }}^{m\leftarrow m^{\prime }}\left(\omega \right)\;=\;L_{n\to n^{\prime }}^{m\leftarrow m^{\prime }}\left(\omega \right)\;+\sum_{n_{1}m_{1}}\sum_{n_{2}m_{2}}L_{n\to n_{1}}^{m\leftarrow m_{1}}\left(\omega \right)\;\Xi_{n_{1}\to n_{2}}^{m_{1}\leftarrow m_{2}}\;L_{n_{2}\to n^{\prime }}^{m_{2}\leftarrow m^{\prime }}\left(\omega \right),$$
where the time-ordered electron–hole propagator is defined as
(23)$$L_{n\to n^{\prime }}^{m\leftarrow m^{\prime }}\left(\omega \right)\;=\delta_{nn^{\prime }}\delta_{mm^{\prime }}\int_{-\infty }^{\infty }\frac{d\omega^{\prime }}{2\pi i}G_{n}\left(\omega^{\prime }\right)G_{m}\left(\omega +\omega^{\prime }\right).$$The propagator (or Green’s function) of an electron or a hole in a molecular state $|\phi_{n}\rangle$ is
(24)$$G_{n}\left(\omega \right)=\frac{1}{\omega -E_{n}+E_{n}^{\mathrm{XC}}-\Sigma_{n}^{X}-\Sigma_{n}^{C,0}\left(\omega \right)},$$
where the exchange of self-energy is
(25)$$\Sigma_{n}^{X}=-\frac{1}{\Omega_{\mathrm{sc}}}\sum_{m}f_{m}\sum_{GG^{\prime }}\rho_{nm}^{*}\left(G\right)V_{GG^{\prime }}^{C}\left(Q\right)\rho_{nm}\left(G^{\prime }\right).$$The correlation self-energy in the G${}_{0}$W${}_{0}$ approximation is
(26)$$\Sigma_{n}^{C,0}\left(\omega \right)=\frac{1}{\Omega_{\mathrm{sc}}}\sum_{m}\sum_{GG^{\prime }}\rho_{nm}^{*}\left(G\right)\rho_{nm}\left(G^{\prime }\right)\left\{\left(1-f_{m}\right)\;\Gamma_{GG^{\prime }}^{0}\left(\omega -E_{m}\right)-f_{m}\;\Gamma_{GG^{\prime }}^{0}\left(E_{m}-\omega \right)\right\},$$
where the correlation propagator is defined as
(27)$$\Gamma_{GG^{\prime }}^{0}\left(\omega \right)=\int_{0}^{\infty }d\omega^{\prime }\frac{S_{GG^{\prime }}^{0}\left(\omega^{\prime }\right)}{\omega -\omega^{\prime }+i\delta },$$
and $f_{n}=\theta \left(E_{F}-E_{n}\right)$ is the Fermi–Dirac distribution at T=0. The spectrum of the electronic excitation in a self-standing molecule is
(28)$$S_{GG^{\prime }}^{0}\left(\omega \right)=-\frac{1}{\pi }ImW_{GG^{\prime }}^{0}\left(\omega \right).$$To avoid double-counting, we excluded the DFT exchange–correlation contribution $E_{n}^{\mathrm{XC}}$ from the KS energy $E_{n}$ in Equation (Equation 24). In the quasi-particle (QP) approximation, the electrons and holes have energies $E_{n}^{\mathrm{QP}}$, which are the real poles of Green’s function (Equation 24), i.e., they satisfy the equation
(29)$$E_{n}-E_{n}^{\mathrm{XC}}+\Sigma_{n}^{X}+Re\Sigma_{n}^{C,0}\left(E_{n}^{\mathrm{QP}}\right)=E_{n}^{\mathrm{QP}}.$$The electron/hole Green functions can now be approximated as
(30)$$\begin{array}{c} G_{n}^{\mathrm{QP}}\left(\omega \right)=\frac{1-f_{n}^{\mathrm{QP}}}{\omega -E_{n}^{\mathrm{QP}}+i\delta }+\frac{f_{n}^{\mathrm{QP}}}{\omega -E_{n}^{\mathrm{QP}}-i\delta }, \end{array}$$
and after they are used in (Equation 23), the time-ordered electron–hole propagator becomes
(31)$$L_{n\to n^{\prime }}^{m\leftarrow m^{\prime }}\left(\omega \right)\;=\frac{f_{n}^{\mathrm{QP}}-f_{m}^{\mathrm{QP}}}{\omega +E_{n}^{\mathrm{QP}}-E_{m}^{\mathrm{QP}}+i\delta sgn\left(E_{m}^{\mathrm{QP}}-E_{n}^{\mathrm{QP}}\right)}\delta_{nn^{\prime }}\delta_{mm^{\prime }},$$
where $\delta =0^{+}$. The ‘time-ordered’ screened Coulomb interaction is the solution of Dyson’s equation
(32)$$W_{GG^{\prime }}^{0}\left(\omega \right)=V_{GG^{\prime }}^{C}+\sum_{G_{1}G_{2}}V_{GG_{1}}^{C}\chi_{G_{1}G_{2}}^{0}\left(\omega \right)W_{G_{2}G^{\prime }}^{0}\left(\omega \right),$$
where the matrix of the ’time-ordered’ irreducible polarizability is
(33)$$\chi_{GG^{\prime }}^{0}\left(\omega \right)=\frac{2}{\Omega_{\mathrm{sc}}}\sum_{nm}\frac{\left(f_{n}-f_{m}\right)\rho_{nm}\left(G\right)\;\rho_{nm}^{*}\left(G^{\prime }\right)}{\hbar \omega +E_{n}-E_{m}+i\delta sgn\left(E_{m}-E_{n}\right)},$$
and the charge vertices are
(34)$$\rho_{nm}\left(G\right)=\int_{\Omega_{\mathrm{sc}}}dr\;\phi_{n}^{*}\left(r\right)e^{-iGr}\phi_{m}\left(r\right).$$Since we calculate the optical conductivity of a single isolated benzene molecule, we have to exclude the effect on its polarizability due to the interaction with the surrounding molecules in the sc lattice. This is accomplished by using the truncated Coulomb interaction [67]
(35)$$V_{C}\left(r-r^{\prime }\right)=\frac{\Theta \left(|r-r^{\prime }|-R_{C}\right)}{|r-r^{\prime }|},$$
where Θ is the Heaviside step function, and $R_{C}$ is the range of the Coulomb interactions, i.e., the radial cutoff. The Coulomb interaction matrix to be used in (Equation 32) is then
(36)$$V_{GG^{\prime }}^{C}=\frac{1}{\Omega }\int_{\Omega }dr\int_{\Omega }dr^{\prime }e^{-iGr}V_{C}\left(r,r^{\prime }\right)e^{iG^{\prime }r^{\prime }}=\frac{4\pi }{{\left|G\right|}^{2}}\left[1-\cos|G\right|R_{C}]\delta_{GG^{\prime }}.$$The Bethe–Salpeter kernel is
(37)$$\Xi_{n\to n^{\prime }}^{m\leftarrow m^{\prime }}\;=\;\Xi_{n\to n^{\prime }}^{H,m\leftarrow m^{\prime }}-\;\frac{1}{2}\Xi_{n\to n^{\prime }}^{F,m\leftarrow m^{\prime }}$$
where the BS–Hartree kernel is
(38)$$\Xi_{n\to n^{\prime }}^{H,m\leftarrow m^{\prime }}\;=\frac{1}{\Omega_{\mathrm{sc}}}\sum_{G_{1}G_{2}}\rho_{nm}^{*}\left(G_{1}\right)\;V_{G_{1}G_{2}}^{C}\rho_{n^{\prime }m^{\prime }}\left(G_{2}\right),$$
and the BS–Fock kernel is
(39)$$\Xi_{n\to n^{\prime }}^{F,m\leftarrow m^{\prime }}\;=\frac{1}{\Omega_{\mathrm{sc}}}\sum_{G_{1}G_{2}}\rho_{nn^{\prime }}^{*}\left(G_{1}\right)W_{G_{1}G_{2}}^{0}\left(\omega =0\right)\rho_{mm^{\prime }}\left(G_{2}\right).$$Here, the index ‘0’ in *W*, *S*, Γ, and $\Sigma^{C}$ emphasizes that we consider the screened interaction in a self-standing molecule. Finally, considering that the interaction $V_{C}$ prevents the correlations between the conductivities in the adjacent cells, the conductivity of an isolated molecule is equal to the conductivity per unit cell
(40)$$\sigma_{\mu \nu }\left(\omega \right)=\int_{\Omega_{\mathrm{sc}}}dr\int_{\Omega_{\mathrm{sc}}}dr^{\prime }\sigma_{\mu \nu }^{3D}\left(r,r^{\prime },\omega \right).$$After using (Equation 19) in (Equation 40), the optical conductivity of a single molecule becomes
(41)$$\sigma_{\mu \nu }\left(\omega \right)\;=\;\Omega_{\mathrm{sc}}\sigma_{G=0G^{\prime }=0}^{\mu \nu ,3D}\left(\omega \right).$$After combining Equations (Equation 9), (Equation 20), and (Equation 41), we determine the explicit expression for the surface optical conductivity
(42)$${\tilde{\sigma }}_{\mu \nu }\left(\omega \right)=\frac{2i}{\omega S_{\mathrm{fcc}}}\sum_{nm}\sum_{n^{\prime }m^{\prime }}j_{nm}^{\mu }\left(G=0\right)K_{n\to n^{\prime }}^{m\leftarrow m^{\prime }}\left(\omega \right){\left[j_{n^{\prime }m^{\prime }}^{\nu }\left(G^{\prime }=0\right)\right]}^{*},$$
which can be used in Dyson’s equation for the electric field propagator (Equation 10). It is important to note that the dimension of the conductivity (Equation 42) is exactly the quantum of conductance $G_{0}=\frac{2\pi e^{2}}{h}$, already the standardized unit for describing the optical conductivity in 2D crystals [28,47,65]. Accordingly, the ${\tilde{\sigma }}_{\mu \nu }\left(\omega \right)$ represents the optical conductivity of one (e.g., *i*-th) molecular layer.

### 2.3. Optical Conductivity in a Molecule Physisorbed at a Dielectric Surface

We assume that a fullerene molecule, centered at $z=z_{i}$, is physisorbed at the supporting crystal occupying the z<0 half-space, as illustrated in Figure 3.

We further assume that the bonding between the molecule and the supporting crystal has the vdW character, which implies a small electronic overlap between the molecule and the substrate and, therefore, a small impact on the short-range electronic correlations to the molecular QP and optical properties. More precisely, the processes involving the electronic hopping between the molecule and the supporting crystal are neglected. However, we shall retain the processes of scattering an electron or a hole or an excited electron–hole pair by the potential induced at the crystal surface, ΔV.

If we have two valence electrons at points r and $r^{\prime }$ in a self-standing molecule, they interact via the bare Coulomb truncated potential $V_{C}$ (Equation 35), but they also polarize the molecule itself so that the total interaction between them is given by the potential $W^{0}$, obtained as the solution of Equation (Equation 32). After the polarizable surface is brought close to the molecule, the electrons at r and $r^{\prime }$ polarize the surface as well so that the interaction between them, neglecting the polarization of the molecule, becomes
(43)$$V_{S}\left(r,r^{\prime },\omega \right)\;=\;V_{C}\left(r,r^{\prime }\right)+\Delta V_{S}\left(r,r^{\prime },\omega \right),$$
where $\Delta V_{S}\left(r,r^{\prime },\omega \right)$ is the induced dynamic Coulomb potential coming from the excitations of the electronic modes or phonons at the surface. The total interaction between the electrons (including the polarization of the molecule) is then the solution of Dyson’s equation
(44)$$W_{GG^{\prime }}^{S}\left(\omega \right)=V_{GG^{\prime }}^{S}\left(\omega \right)+\sum_{G_{1}G_{2}}V_{GG_{1}}^{S}\left(\omega \right)\chi_{G_{1}G_{2}}^{0}\left(\omega \right)W_{G_{2}G^{\prime }}^{S}\left(\omega \right),$$
where
(45)$$\Delta V_{GG^{\prime }}^{S}\left(\omega \right)=\frac{1}{\Omega_{\mathrm{sc}}}\int_{\Omega_{\mathrm{sc}}}dr\int_{\Omega_{\mathrm{sc}}}dr^{\prime }e^{-iGr}\Delta V_{S}\left(r,r^{\prime },\omega \right)e^{iG^{\prime }r^{\prime }}.$$Here, the integration is constrained within the unit cell $\Omega_{\mathrm{sc}}$ centered at $r_{0}=\left(0,0,z_{i}\right)$ (to avoid interaction with the neighboring molecules, via $\Delta V_{S}$), as shown in Figure 3. The induced interaction in the region z>0 can be written as [62]
(46)$$\Delta V\left(r,r^{\prime },\omega \right)=\int \frac{dQ}{{\left(2\pi \right)}^{2}}v_{Q}D\left(Q,\omega \right)e^{iQ(\rho -\rho^{\prime })}e^{-Q(z+z^{\prime })},$$
where $v_{Q}=\frac{2\pi }{\left|Q\right|}$. Since the supporting crystal dielectric response is approximated by the local 3D macroscopic dielectric function $\epsilon_{M}\left(\omega \right)$, the surface excitation propagator can be approximated as [68]
(47)$$D\left(Q,\omega \right)\approx D\left(\omega \right)=\frac{1-\epsilon_{M}\left(\omega \right)}{1+\epsilon_{M}\left(\omega \right)}.$$After (Equation 46) and (Equation 47) are used in (Equation 45), we have
(48)$$\Delta V_{GG^{\prime }}^{S}\left(\omega \right)=\frac{D(\omega )}{\Omega_{\mathrm{sc}}}\int \frac{dQ}{{\left(2\pi \right)}^{2}}v_{Q}e^{-2Qz_{i}}F_{G}\left(Q\right)F_{G^{\prime }}^{*}\left(Q\right),$$
where the form factors are defined as
$$F_{G}\left(Q\right)=8{\left(-1\right)}^{n_{z}}\frac{\sin\left[\left(Q_{x}-G_{x}\right)\frac{a}{2}\right]\sin\left[\left(Q_{y}-G_{y}\right)\frac{a}{2}\right]\sinh\left[Q\frac{a}{2}\right]}{\left(Q_{x}-G_{x}\right)\left(Q_{y}-G_{y}\right)\left(Q+iG_{z}\right)}.$$The reciprocal vectors of the sc lattice are $G=(G_{x},G_{y},G_{z})$, where $G_{x}=\frac{2\pi n_{x}}{a}$, $G_{y}=\frac{2\pi n_{y}}{a}$, $G_{z}=\frac{2\pi n_{z}}{a}$ and $n_{x},n_{y},n_{z}\in Z$. After we determine the ‘bare’ potential (which includes the polarization of the surface) $V_{S}$ and the ‘total’ potential (which includes the polarization of the surface and of the molecule) $W_{S}$, the calculation of the QP and optical properties of a molecule near the dielectric surface is equal to the procedure described in Section 2.2, except for that in the BS–Hartree kernel (Equation 38), we need to replace
$$V_{G_{1}G_{2}}^{C}\to V_{G_{1}G_{2}}^{S}\left(\omega \right),$$
in the BS–Fock kernel (Equation 39)
$$W_{G_{1}G_{2}}^{0}\left(\omega =0\right)\to W_{G_{1}G_{2}}^{S}\left(\omega =0\right),$$
and the spectrum of the electronic excitations (Equation 28) used to calculate the correlation self-energy (Equation 26) is
(49)$$S_{GG^{\prime }}^{0}\left(\omega \right)\to S_{GG^{\prime }}^{S}\left(\omega \right)=-\frac{1}{\pi }ImW_{GG^{\prime }}^{S}\left(\omega \right).$$

### 2.4. Computational Details

The fullerene KS orbitals $\phi_{n}\left(r\right)$ and energies $E_{n}$ were calculated using the plane-wave self-consistent field DFT code (PWSCF) within the QUANTUM ESPRESSO (QE) package [69]. The core-electron interaction was approximated by the norm-conserving pseudopotentials [70,71] so that the number of occupied valance states was 120. The exchange-correlation (XC) potential was approximated by the Perdew–Burke–Ernzerhof (PBE) generalized gradient approximation (GGA) functional [72]. The plane-wave cut-off energy was 60 Ry. The molecules were arranged in the simple cubic Bravais lattice of the unit cell a=18 Å with one molecule per unit cell. Since there was no intermolecular overlap, the ground state electronic density was calculated at the Γ point only. The geometries were fully relaxed, with all forces ≲0.02 eV/Å. The RPA ’time-ordered’ screened Coulomb interactions $W^{0,S}$ (Equations (Equation 32) and (Equation 44)) were calculated using the energy cut-off 2 Ry (∼27 eV), and the band summations ‘(n,m)’ in the irreducible polarizability (Equation 33) were performed over 240 molecular states. The exchange self-energy (Equation 25) was calculated using the energy cut-off 8 Ry (∼109 eV) and the correlation self-energy (Equation 26) was determined (according to the cut-off in $W_{0}$) using the energy cut-off 2 Ry (∼27 eV); the band summation ‘*m*’ was performed using 240 molecular orbitals. The BS–Hartree kernel (Equation 38) and the ‘bare’ BS–Fock kernel (the Equation (Equation 39), derived using the bare interaction $V_{C}$), were calculated using the energy cut-off 8 Ry (∼109 eV); the induced Fock kernel (the Equation (Equation 39), derived using the induced interaction $W_{0,S}-V_{C}$), was calculated using the energy cut-off 2 Ry (∼27 eV). During the evaluation of the BSE–HF kernels, we used 42 occupied (HOMO − 41, …, HOMO) and 42 unoccupied (LUMO, …, LUMO + 41) molecular states, i.e., the dimensions of the BSE–HF kernel matrix was 2×42×42=3528. To achieve the accurate (experimental) exciton energy, the calculation was performed beyond the Tamm–Dancoff approximation. The damping parameters δ used in (Equation 31) and in (Equation 33) were 50 meV and 200 meV, respectively. We assume that the dielectric medium was vacuum (i.e., $\epsilon_{0}=1$), and that the substrate was the aluminum-oxide Al${}_{2}$O${}_{3}$, described by the macroscopic dielectric function
(50)$$\epsilon_{M}\left(\omega \right)=1/\epsilon_{G=0G^{\prime }=0}^{-1}\left(q\approx 0,\omega \right),$$
where the dielectric matrix is $\hat{\epsilon }=\hat{I}-\hat{V}{\hat{\chi }}^{0}$. The irreducible polarizability $\chi^{0}$ is determined using Equation (Equation 33) for $\Omega_{\mathrm{sc}}\to \Omega$, n→(n,k) and m→(m,k+q). Here k, q, and G are the 3D wave vector, the transfer wave vector, and the reciprocal lattice vector, respectively, corresponding to the bulk Al${}_{2}$O${}_{3}$ crystal. The bare Coulomb interaction is $V_{GG^{\prime }}\left(q\right)=\frac{4\pi }{{\left|q+G\right|}^{2}}\;\delta_{GG^{\prime }}$. The ground state electronic density of the bulk Al${}_{2}$O${}_{3}$ is calculated using 9×9×3 K-mesh, the plane-wave cut-off energy is 50 Ry, and the Bravais lattices are hexagonal (12 Al and 18 O atoms in the unit cell) with the lattice constants $a_{\mathrm{Al}}{}_{2}O{}_{3}=4.76$ Å and $c_{\mathrm{Al}}{}_{2}O{}_{3}=12.99$ Å. The Al${}_{2}$O${}_{3}$ irreducible polarizability $\chi^{0}$ is calculated using the 21×21×7*k*-point mesh and the band summations (n,m) are performed over 120 bands. The damping parameter is δ=100 meV and the temperature is T=10 meV. For the optically small wave vectors q≈0, the crystal local field effects are negligible, i.e., the crystal local field effects cut-off energy is set to zero. Using this approach, the Re$\epsilon_{M}$ is almost constant (Re$\epsilon_{M}\approx 3$) for low frequencies (ω<3 eV), i.e., in the IR and visible range, while Im$\epsilon_{M}$ is equal to zero up to the band gap energy ($E_{g}$∼6 eV). Therefore, Al${}_{2}$O${}_{3}$ is a good choice for the substrate in the visible and near-UV frequency range, since its electronic excitations are above that range, and its IR active SO phonons (at $\omega_{\mathrm{SO}}<100$ meV) [73] are far below the C${}_{60}$ excitons, which means that in the frequency range of interest, there is no dissipation of the electromagnetic energy in the substrate (it is transparent). In addition to that, the dielectric function is mostly constant, but in this calculation, we used the fully dynamical and complex $\epsilon_{M}\left(\omega \right)$. The integration in (Equation 48) was performed over the two-dimensional wave vectors $Q=(Q_{x},Q_{x})$ using a 121×121 rectangular mesh and the cut-off wave vector $Q_{C}=0.5$ a.u. For the radial cut-off in the truncated interaction (Equation 36), we used $R_{C}=a/2=9$ Å.

## 3. Results and Discussion

Strong exciton–photon hybridization usually occurs due to the large overlap between excitons of large oscillatory strengths and confined photons. This is experimentally achieved by placing nanostructures or molecules of large oscillatory strength in a metallic cavity. Our goal here was to explore the electromagnetic eigenmodes, which are a mixture of ‘free’ s(TE) photons and excitons, i.e., a ‘free’ traveling photon (not confined between metallic walls) captured by an exciton and ‘trapped’ in a molecular nanostructure. This concept makes it possible to clearly estimate the contribution of the ‘free’ photon in the hybrid exciton–photon mode.

First, we determined the QP and optical properties of a self-standing layer of the $C_{60}$ molecules, which corresponded with the properties of the gas-phase molecule, since at the $G_{0}W_{0}$-BSE stage of the calculations, the molecules were not coupled. The calculated G${}_{0}$W${}_{0}$ HOMO-LUMO gap was $E_{g}=4.66$ eV, which is in good agreement with the experimental value 5 eV [74,75,76,77]. When a molecular layer is deposited on an Al${}_{2}$O${}_{3}$ dielectric surface, the band gap reduces to $E_{g}=4.26$ eV. Here, the separation was chosen to be $z_{0}=6.5$ Å which corresponds to the characteristic vdW atom–atom separation of 3 Å (Note that $z_{0}$ is defined as the distance between the substrate and the center of the molecule, as denoted in Figure 1, so the C${}_{60}$ molecule radius (3.57 Å) has to be added to the vdW atom–atom distance). For comparison, we determined the HOMO-LUMO gap for the molecule at a silver (Ag) surface also described in terms of the *ab initio* macroscopic dielectric function. For the same separation ($z_{0}=6.5$ Å), the gap was $E_{g}=3.81$ eV; compared with the insulator surface, the reduction is twice as strong. The image theory estimation of the HOMO-LUMO gap at the metallic surface is $E_{g}$∼3.56 eV which, as expected, overestimates the G${}_{0}$W${}_{0}$ result. Figure 4a shows the calculated G${}_{0}$W${}_{0}$-BSE optical conductivities ${\tilde{\sigma }}_{x}\left(\omega \right)$ in the self-standing molecular layer (black solid) and in the molecular layer deposited at the Al${}_{2}$O${}_{3}$ dielectric surface (cyan dashed). For comparison, the experimental optical absorption in the fullerene C${}_{60}$ [78] is presented by the red circles. This experimental study, as well as some others [79,80], show three broad excitation bands at $\omega_{ex1}$∼ 3.9 eV, $\omega_{ex2}$∼4.9 eV and $\omega_{ex3}$∼6 eV, which is in very good agreement with our results. It should be emphasized that, since the band gap is $E_{g}=4.66$ eV, only the excitation band $\omega_{ex1}$ can be considered as an exciton by definition, and its binding energy is $E_{g}-\hbar \omega_{ex1}$∼0.76 eV. In the literature, the strong maximum $\omega_{ex3}$ is usually referred to as the π plasmon [81,82]. All three excitons are the result of the electronic transitions within the $C_{60}$π-complex [83]. When the molecule is deposited at the Al${}_{2}$O${}_{3}$ surface, the excitation band barely changes at all, due to the well-known cancellation effect: the substrate weakens the interaction between the excited electron and the hole, which reduces the exciton binding energy and, therefore, cancels the gap reduction. This phenomenon was studied in detail in references [62,66,84]. Since the influence of the dielectric surface on the molecular optical conductivity is weak, we shall further approximate ${\tilde{\sigma }}_{i}^{x}\left(\omega \right)\approx {\tilde{\sigma }}^{x}\left(\omega \right)$, where ${\tilde{\sigma }}^{x}\left(\omega \right)$ is the optical conductivity in the self-standing molecular single layer, i.e., for $z_{0}\to \infty$.

This means that the impact of the dielectric substrate on the electromagnetic modes in the $C_{60}$ films is reduced to the propagator $\Gamma^{\mathrm{sc}}$, which appears in Dyson’s Equation (Equation 14). The spectra of the s(TE)-polarized electromagnetic modes in the $C_{60}$ films will be analyzed using the real part of the propagator E in the topmost molecular layer
(51)$$S\left(Q,\omega \right)\;=\;ReE_{xx}(|Q|y,\omega ,z_{N},z_{N}).$$As already mentioned, the spacing parameters were chosen to be $z_{0}=6.5$ Å and Δ=8.1 Å. In the near field spectroscopy experiment, the incident photon of the wavelength
(52)$$\lambda_{\mathrm{ex}1}=\frac{2\pi c}{\omega_{\mathrm{ex}1}\sqrt{\epsilon (\omega_{\mathrm{ex}1})}}$$
couples to polarization modes in some sub-wavelength nanostructure (e.g., AFM tip) and is scattered (diffracted) to all $\lambda >\lambda_{\mathrm{ex}1}$ and $\lambda <\lambda_{\mathrm{ex}1}$ so that it could excite all radiative electromagnetic modes, as well as all evanescent ones at a fixed frequency $\omega_{\mathrm{ex}1}$. The Rabi splitting (the measure of the free-photon participation in the exciton–polariton mode) is defined as the difference between the exciton $\omega_{ex1}$ and the exciton–polariton $\omega_{\mathrm{ex}1-\mathrm{pol}}$
(53)$$\hbar \Omega =\hbar \omega_{\mathrm{ex}1}-\hbar \omega_{\mathrm{ex}1-\mathrm{pol}}\left(Q_{ex1}\right),$$
where $Q_{ex1}=2\pi /\lambda_{\mathrm{ex}1}$ is the wave vector at which the exciton $\omega_{\mathrm{ex}1}$ and the photon cross.

Figure 4b shows the dispersion relations of the exciton–polaritons $\omega_{\mathrm{ex}1-\mathrm{pol}}\left(Q\right)$ in the self-standing $C_{60}$ film (blue dots), and in the $C_{60}$ film deposited at a Al${}_{2}$O${}_{3}$ surface (red dots). The number of $C_{60}$ single layers in the film was chosen to be N=6. The energies of the exciton–polaritons $\omega_{\mathrm{ex}1-\mathrm{pol}}\left(Q\right)$ correspond to the maxima in the spectral function S(Q,ω), appearing below the exciton energy $\omega_{\mathrm{ex}1}$ [see Figure 4c]. For the self-standing film, ϵ(ω)=1, so that the crossing wave vector, according to the Equation (Equation 52), was $Q_{\mathrm{ex}1}^{0}=0.02\;{\mathrm{nm}}^{-1}$, and the Rabi splitting, according to the Equation (Equation 53), was $\hbar \Omega_{0}=650$ meV. The red dots in Figure 4b show that the presence of the substrate reduces the bending of the exciton–polariton dispersion relation significantly, therefore reducing the corresponding Rabi splitting. In this case, $\epsilon \left(\omega \right)=\epsilon_{M}\left(\omega \right)$, which gives the crossing wave vector $Q_{\mathrm{ex}1}^{S}=0.035\;{\mathrm{nm}}^{-1}$ and, thus, the Rabi splitting $\hbar \Omega_{S}=103$ meV.

Figure 4c shows the spectra of the s(TE)-polarized electromagnetic modes $S(Q_{ex1},\omega )$ in the self-standing $C_{60}$ films for $Q_{\mathrm{ex}1}^{0}=0.02\;{\mathrm{nm}}^{-1}$ (blue solid) and in the $C_{60}$ films deposited at the Al${}_{2}$O${}_{3}$ surface for $Q_{\mathrm{ex}1}^{S}=0.035\;{\mathrm{nm}}^{-1}$ (red dashed). The number of layers is set to be N=0,1,2,3,⋯,10, where the case N=0 corresponds to the spectra of the free photons in vacuum or at the vacuum/Al${}_{2}$O${}_{3}$ interface (photons continuum). Due to the lower intensity, all spectra for the supported films were multiplied by factor 2. We can clearly see the exciton–polariton peaks separating from the photon continuum as the number of layers increases. In the self-standing films, the exciton–polariton is already well separated for N=2, while in the supported films, it occurs for N=4. For N=10, the giant light–matter coupling was achieved providing large Rabi splitting $\Omega_{0}=1334$ meV and $\Omega_{S}=670$ meV in the self-standing and the supported films, respectively. Here we limited our study to the exciton–polaritons in the quasi-2D nanostructures so that the maximum number of the molecular layers was limited to N=10, still satisfying the sub-wavelength limit $\lambda_{\mathrm{ex}1}>\left(N-1\right)\Delta$. For a further increase of the number of layers *N*, the Rabi splitting continued to increase, but the experimental limitations in the realization of such a system raise the question of the plausibility of such a result.

We need to emphasize that the anti-crossing behavior and the Rabi splitting into the lower polariton band (LPB) and the upper polariton band (UPB) is a well-defined concept only when describing the interaction between the well-defined eigenmodes, such as excitons and cavity photons [33] or BSWs [48]. In our case, the only well-defined modes (boson) were the excitons $\omega_{\mathrm{ex}1}$-$\omega_{\mathrm{ex}3}$. On the other hand, the photons appeared as a continuum of eigenmodes above the light lines Qc or $Qc/\sqrt{\epsilon_{M}}$. This means that it only made sense to observe the deformation of the exciton dispersion $\omega_{\mathrm{ex}1}$, characterized by the exciton bending parameter Ω. However, in Figure 5, it seems that the edge of the photon continuum Qc or $Qc/\sqrt{\epsilon_{M}}$ behaves similar to a well-defined eigenmode, so in Figure 4b,c we also introduced and denoted the LPB and UPB, exactly as they appear in Figure 5b,e. The LPB corresponds to the exciton–polariton dispersion relation $\omega_{\mathrm{ex}1-\mathrm{pol}}\left(Q\right)$ appearing in (Equation 53).

To present a more extensive picture of the electromagnetic modes in the $C_{60}$ films, including the hybridizations of the photon and the higher excitons $\omega_{\mathrm{ex}2}$ and $\omega_{\mathrm{ex}3}$, Figure 5a–c show the spectral intensity of the s(TE)-polarized electromagnetic modes S(Q,ω) in the self-standing $C_{60}$ films for N=3, 6, and 9, respectively. Figure 5d–f show the same for the $C_{60}$ films deposited at the Al${}_{2}$O${}_{3}$ surface. It is immediately obvious that the most intensive electromagnetic modes occur in the evanescent regions ω<Qc and $\omega <Qc/\epsilon_{M}\left(\omega \right)$, for the self-standing and supported films, respectively. These modes produce an electric field, which spreads within the film and exponentially decays outside of the film [as illustrated in Figure 2b], which is one of the inherent properties of the 2D exciton–polaritons. Moreover, as the field is more confined, it is stronger, so that the polariton modes confined at a sub-wavelength scale are very interesting for many applications. The free photon continua can be seen as the low-intensity patterns in the regions ω>Qc, and $\omega >Qc/\epsilon_{M}\left(\omega \right)$ in the self-standing and supported films, respectively, with their intensity weakening with the number of layers *N*. If the electromagnetic eigenmode, for example in the $C_{60}$ film at the Al${}_{2}$O${}_{3}$, would occur in the region $\omega >Qc/\epsilon_{M}\left(\omega \right)$, it would be irradiated (decay radiatively) into the Al${}_{2}$O${}_{3}$ crystal, so that these modes appear only as weak resonances. Such weak resonances can be seen as weak horizontal patterns (parallel with the excitons $\omega_{\mathrm{ex}1}$-$\omega_{\mathrm{ex}3}$) entering the radiative region.

In Figure 5a, we can see the beginning of the hybridization between the free photon and the three $C_{60}$ excitons ($\omega_{\mathrm{ex}1}$-$\omega_{\mathrm{ex}3}$), which visibly intersect the photon line ω=Qc. In Figure 5b,c, the photon line is already significantly deformed between the excitons and pushed far into the evanescent region, which means that the photon Qc and the three excitons $\omega_{\mathrm{ex}1}$, $\omega_{\mathrm{ex}2}$, and $\omega_{\mathrm{ex}3}$ are strongly coupled and converted into three exciton–polaritons $\omega_{\mathrm{ex}1-\mathrm{pol}}$, $\omega_{\mathrm{ex}2-\mathrm{pol}}$, and $\omega_{\mathrm{ex}3-\mathrm{pol}}$. In Figure 5d–f, the exciton–polaritons are less intense and pushed into the evanescent region $\omega <Qc/\epsilon_{M}\left(\omega \right)$, the use of the substrate obviously weakens the intensity of the exciton–polaritons. However, in Figure 5e,f, we can see the formation of the three strong exciton–polariton modes $\omega_{\mathrm{ex}1-\mathrm{pol}}$–$\omega_{\mathrm{ex}3-\mathrm{pol}}$, for larger numbers of layers N=6 and 9, respectively. Finally, this confirms that the strong binding between the transverse s(TE) photon and the molecular excitons is a realistic physical phenomenon that can be achieved experimentally. The use of a metal substrate is not recommended because a metallic surface will strongly quench the exciton–polariton modes. To support this argument, we performed the same calculation using a silver (Ag) substrate and obtained much weaker exciton–polariton modes.

Finally, we noticed that the hybridization between the exciton and the photon increases weakly if just the exciton oscillatory strength (e.g., S) increases, but strongly if the number of spatially separated molecular layers N increases. For example, the exciton–photon binding is much stronger in the N-separated layers of the oscillatory strength S than in one layer of the oscillatory strength N× S. This suggests that the photon-exciton coupling can be increased simply by increasing the number of layers N in a multilayered van der Waals heterostructure.

## 4. Conclusions

We showed that the 2D-layered heterostructures, consisting of a larger number of exciton-active single layers or 2D crystals (e.g., N>5), can support evanescent s(TE)-polarized exciton–polaritons with strong photonic character. We obtained giant Rabi splitting of more than $\Omega_{0}=1000$ meV and $\Omega_{S}=500$ meV in the self-standing and supported C${}_{60}$ films for N=10, respectively. This investigation has fundamental and practical contributions. We predict the existence of the evanescent s(TE) polarization modes with significant photonic character, which vanish exactly without the photon admixture (for c→∞). Unlike the well-known p(TM) polarization modes (e.g., the plasmon polariton for c→∞ collapses into a plasmon), this is a new fundamental contribution. We also demonstrated that exciton–photon coupling can be manipulated simply by changing the number of single layers (*N*) in a vdW-layered heterostructure. Moreover, due to the fact that the vdW heterostructure with a thickness of just a few molecular (or atomic) layers supports the confined photons, it can be easily implemented in the photonic integrated circuits or photonics chips. The disadvantage of these types of photonic modes is that they do not couple directly to the free photons (external light). However, once excited, these modes can be easily manipulated (since they are trapped in the nanostructure). For example, by patterning the vdW nanoribbons on a dielectric wafer (patterning the photonics circuits), the direction of the photon propagation can easily be modified. Moreover, the exciton–polaritons can be easily switched (at the contact) into evanescent modes in another nanostructure. Finally, these layered vdW heterostructures can be applied in photonic devices, such as light sources (LED), telecommunications (as waveguides or optical cables), or chemical and biological sensing.

## Figures and Tables

**Figure 1 ijms-23-06943-f001:**
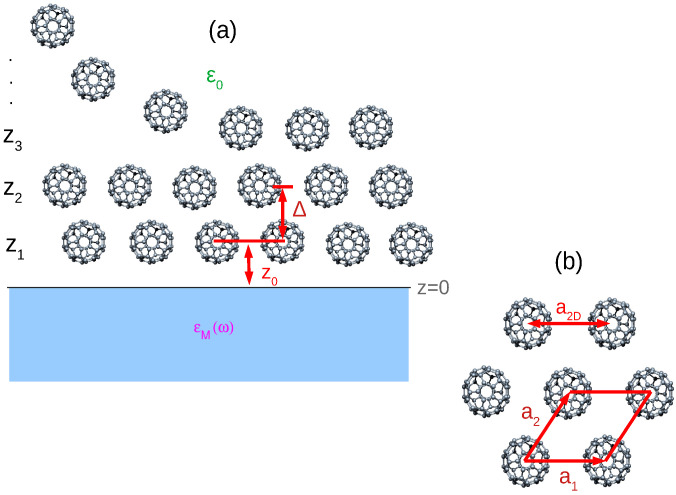
(**a**) C${}_{60}$ molecules upon deposition on the surface self-assemble in a regular FCC structure forming a (111) surface. The C${}_{60}$ layers, occupying z>0 half-space, are immersed in a dielectric medium described by a dielectric constant $\epsilon_{0}$. The dielectric response of the substrate, occupying z<0 half-space, is approximated by a local macroscopic dielectric function $\epsilon_{M}\left(\omega \right)$. The FCC lattice constant is $a_{3D}=14$ Å so that the separation between the layers is $\Delta =a_{3D}/\sqrt{3}=8.1$ Å. (**b**) Each crystal plane forms a 2D-hexagonal Bravais lattice with lattice constant $a_{2D}=a_{3D}/\sqrt{2}=9.9$ Å.

**Figure 2 ijms-23-06943-f002:**
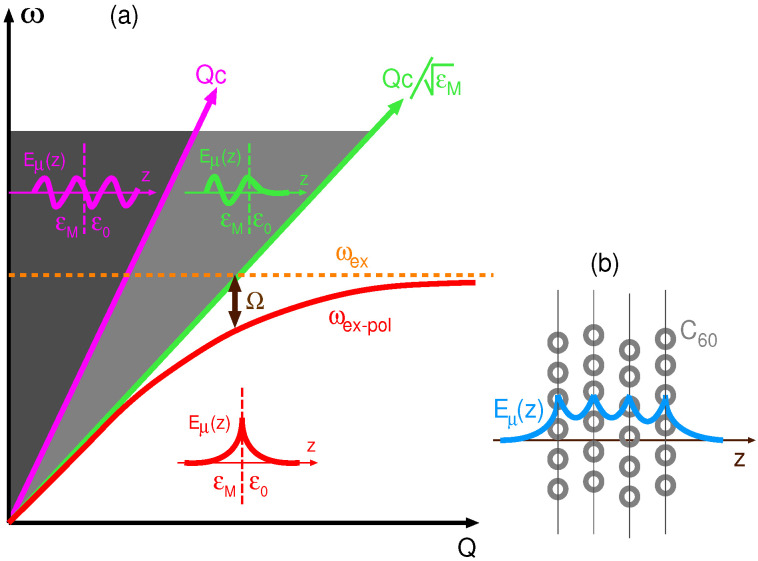
(**a**) The character of the electromagnetic modes at the dielectric/vacuum ($\epsilon_{M}/\epsilon_{0}$) interface. In the region ω>Qc, the electromagnetic modes are entirely radiative (both in vacuum and in the dielectric), in the region $Qc/\sqrt{\epsilon_{M}}<\omega <Qc$, they are radiative in the dielectric and evanescent in vacuum, and in the $\omega <Qc/\sqrt{\epsilon_{M}}$ region, they have fully evanescent character. In the latter region, the photon and molecular exciton ($\omega_{\mathrm{ex}}$) hybridize, and an exciton–polariton ($\omega_{\mathrm{ex}-\mathrm{pol}}$) occurs. The measure of the coupling strength between the exciton $\omega_{\mathrm{ex}}$ and the photon is given by Rabi splitting Ω. (**b**) The evanescent electric field $E_{\mu }\left(z\right)$ produced by an exciton–polariton in the $C_{60}$ film.

**Figure 3 ijms-23-06943-f003:**
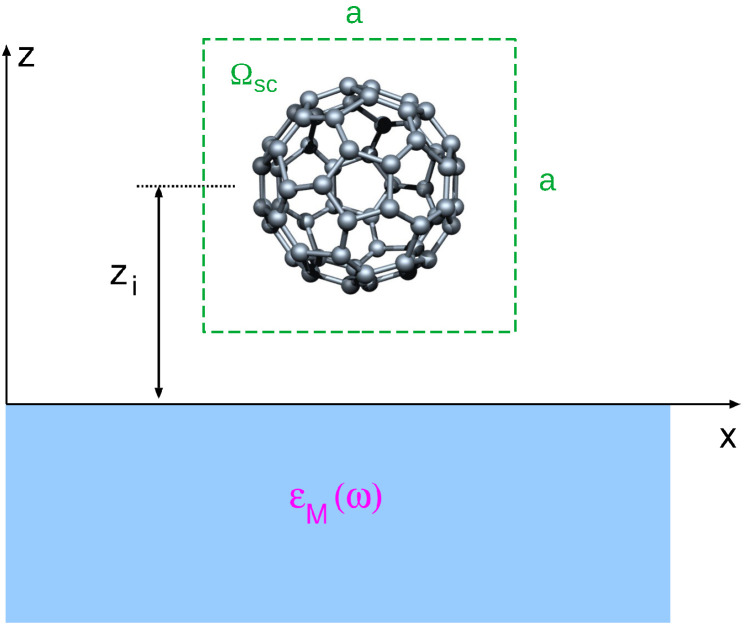
The fullerene molecule C${}_{60}$ centered at $z=z_{i}$ is physisorbed at the supporting crystal occupying the half-space z<0, with the dielectric properties approximated by the macroscopic dielectric function $\epsilon_{M}\left(\omega \right)$.

**Figure 4 ijms-23-06943-f004:**
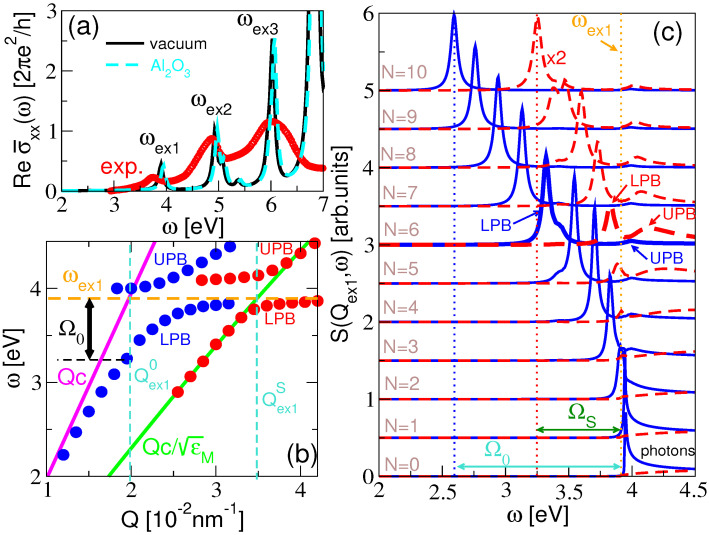
(**a**) The optical conductivities ${\tilde{\sigma }}_{xx}\left(\omega \right)$ in the $C_{60}$ single layer in vacuum (black solid), in the $C_{60}$ single layer at the Al${}_{2}$O${}_{3}$ dielectric surface (cyan dashed) (where $z_{0}=6.5$ Å), and the experimental optical absorption in the gas phase fullerene C${}_{60}$ (red circles). (**b**) The lower and upper exciton–polariton branches, LPB and UPB, respectively, in the self-standing $C_{60}$ film (blue dots) and the $C_{60}$ film deposited at the Al${}_{2}$O${}_{3}$ surface (red dots). The LPB corresponds to the dispersion relation of the exciton–polariton $\omega_{\mathrm{ex}1-\mathrm{pol}}\left(Q\right)$ appearing in Equation (Equation 53). The number of $C_{60}$ layers is N=6. (**c**) The spectra of the s(TE)-polarized electromagnetic modes $S(Q_{ex1},\omega )$ in the self-standing $C_{60}$ films for $Q_{\mathrm{ex}1}^{0}=0.02\;{\mathrm{nm}}^{-1}$ (blue solid) and in the $C_{60}$ films at the Al${}_{2}$O${}_{3}$ surface for $Q_{\mathrm{ex}1}=0.035\;{\mathrm{nm}}^{-1}$ (red dashed). The number of the single layers is N=0,1,2,3,⋯,10, where the case N=0 corresponds to the spectrum of the free photons in vacuum or at the vacuum/Al${}_{2}$O${}_{3}$ interface (the photon continuum). All spectra for the supported films are multiplied by factor 2.

**Figure 5 ijms-23-06943-f005:**
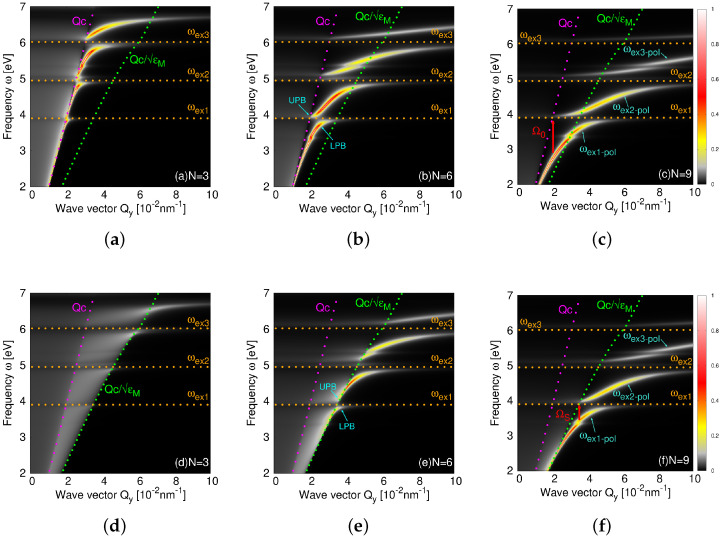
The spectral intensity of the s(TE)-polarized electromagnetic modes S(Q,ω) in the self-standing $C_{60}$ films for (**a**) N=3, (**b**) N=6, and (**c**) N=9. Figures (**d**–**f**) show the same for the $C_{60}$ films deposited at the Al${}_{2}$O${}_{3}$ surface. In both cases, the strong electromagnetic modes ($\omega_{\mathrm{ex}1-\mathrm{pol}}$, $\omega_{\mathrm{ex}2-\mathrm{pol}}$, and $\omega_{\mathrm{ex}3-\mathrm{pol}}$) occur in the evanescent regions ω<Qc and $\omega <Qc/\epsilon_{M}\left(\omega \right)$.

## Data Availability

Not applicable.

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
