# Peer review of "Prediction of Strong Transversal s(TE) Exciton–Polaritons in C60 Thin Crystalline Films"

_ijms, 2022, doi:10.3390/ijms23136943_

Round 1

Reviewer 1 Report

In the manuscript the authors proved that the efficient light-matter coupling can occur beyond the microcavity device setup, i.e. between ’free’ s(TE) photons and excitons. Furthermore, the results showed that the calculated Rabi splitting is more than W = 500meV for N = 10, with a tendency to increase with N, indicating a very strong photonic character of the exciton-polariton. From my perspective, the manuscript has been carefully thought out and prepared, however, a few issues require additional comment:

11. In the introduction, the authors gave several concepts of microcavities, however, they overlooked new concepts of such devices published in recent years by several teams: Sci Rep 11, 74 (2021); Advanced Materials 31.24 (2019): 1900231; Optica 9.4 (2022): 445-450.

22. The results in figure 5 do not correspond to those in figure 4. For better visualization, the results should be done for the same N values.

33. What’s the role of surface plasmon polaritons (SPPs) and gap surface plasmons (GSPs) in coupling? Please comment the strong modal hybridization and energy transfer within the system. How can you further enhance the interplay of these mods?

44. The authors should carefully describe the potential use of the results in photonic devices. It is true that they listed the applications very generally in the summary, but it is not sufficient, especially in the case of theoretical work.

Author Response

In the manuscript the authors proved that the efficient light-matter coupling can occur beyond the microcavity device setup, i.e. between ’free’ s(TE) photons and excitons. Furthermore, the results showed that the calculated Rabi splitting is more than W = 500meV for N = 10, with a tendency to increase with N, indicating a very strong photonic character of the exciton-polariton. From my perspective, the manuscript has been carefully thought out and prepared, however, a few issues require additional comment:

Authors response:

We thank the referee for the positive review and useful comments. Below we try to respond to all referee’s comments.

1. In the introduction, the authors gave several concepts of microcavities, however, they overlooked new concepts of such devices published in recent years by several teams: Sci Rep 11, 74 (2021); Advanced Materials 31.24 (2019): 1900231; Optica 9.4 (2022): 445-450.

Authors response:

We thank the referee to pointing out these important contributions. The second referee had a very similar comment, so we introduced the 2nd paragraph in the Introduction explaining the mentioned recent research.

2. The results in figure 5 do not correspond to those in figure 4. For better visualization, the results should be done for the same N values.

Authors response:

Figures 4 and 5, although showing the same phenomenon, have different purposes. Figure 4 aims to show the separation of the exciton polaritons from the photon continuum as a function of the number of the molecular layers N, for particular wave vector Q. Figure 5 show the dispersion relations of the exciton-polaritons as function of Q. Therefore, comparing two images for the same number of layers would be redundant information, so 5 we prefer to show the results for different number N. Also, for N = 2 the binding becomes very weak and would not be visible in Fig.5. On the other hand, a comparison can be made for the representative case N = 6 when the binding is already very strong. However, for the sake of completeness in Figure 10, we presented the results for the spectra for a denser number of layers N = 0,1,2,3,4 ..., 10.

3. What’s the role of surface plasmon polaritons (SPPs) and gap surface plasmons (GSPs) in coupling? Please comment the strong modal hybridization and energy transfer within the system. How can you further enhance the interplay of these mods?

Authors response:

We thank the referee for this constructive comment. Our first aim was to explore the hybridization between the excitons and the p(TM) surface or gap plasmon polaritons SPP or GPP. However, the SPP or GPP are already plasmon-photon hybrid modes so their hybridization with excitons provides plasmon-photon-excitons mixture sometimes called plexcitons. However, we switched our goal to explore the electromagnetic eigen-modes which are mixture of the ‘free’ s(TE) photons and excitons which makes it possible to clearly estimate the contribution of the ‘free’ photon in the hybrid exciton-photon mode. We can say that we explore the confined photons which are captured by the excitons. Strong exciton-photon hybridization usually occurs due to the large overlap between the excitons of large oscillatory strength and the confined photon. This is experimentally achieved by placing the nanostructures or molecules of large oscillatory strength in the metallic cavity. Here we explore a different concept of binding. The ‘free’ traveling photon (not confined between the metallic walls) is captured by the exciton and ‘trapped’ in the molecular nano-structure. Furthermore, it is interesting that the hybridization between the exciton and photon increases very weakly if just the exciton oscillatory strength is increased, but strongly if the number of spatially separated molecular layers N is increased. For example, the exciton photon binding is much stronger in the N separated layers of oscillatory strength S then in one layer of oscillatory strength N ∙ S. This suggests that the photon-exciton coupling can be increased simply by increasing the number of layers N in multilayered van der Waals heterostructure. To explain this, we added first and last paragraph in Sec.3 Results and discussion.

4. The authors should carefully describe the potential use of the results in photonic devices. It is true that they listed the applications very generally in the summary, but it is not sufficient, especially in the case of theoretical work.

Authors response:

So far there is no evidence of strong s(TE) photons confined within a few atoms thick vdW heterostructures. Due to the fact that a vdW heterostructure of thickness of just few molecular (or atomic) layers supports the confined photons, it can be easily implemented in the photonic integrated circuits or in photonics chips. The disadvantage of this type of photonic modes is that they do not couple directly to free photons (external light). However, once excited these modes can be easily manipulated (since they are trapped in a nanostructure). For example, by patterning the vdW nanoribbons on a dielectric wafer (patterning the photonics circuits), the direction of photon propagation can be easily manipulated. Also, exciton-polaritons can be easily switched (at the contact) to an evanescent mode in another nanostructure. Moreover, this layered vdW heterostructures can be applied in photonics devices such as light sources (LED) or photodetectors of chemical or biological compounds. We have rewritten the Sec. Conclusions in order to describe this.

Reviewer 2 Report

See attached

Author Response

In the manuscript titled “Prediction of strong transversal s(TE) exciton-polaritons in C60 thin crystalline films”, the authors calculated the optical conductivity and spectra for C60 single layer and two-dimensional (2D) crystal in vacuum or dielectric surface, predicted the existence of s(TE) exciton-polariton. These results have fundamental and practical contributions in photonics and optical devices, and I recommend the publication of this paper to the journal of International Journal of Molecular Sciences after some revision. Detailed comments and questions are as following,

Authors response:

We thank the referee for the positive review and the useful comments. We corrected the manuscript in accordance with the referee’s suggestions as much as possible, and we did our best to improve the English and correct the minor mistakes. Below we try to respond to all the referee’s comments.

1. The typo s(TM) appears many times, for example, in “abstract”, second paragraph on Page 2, and “Conclusions”.

Authors response:

We thank the referee for pointing this out. We have corrected the typo in the new version of the manuscript.

2. The symbol ? is used for both Rabi splitting (for example, in “abstract”) and for space of volume (for example, Page 3), please check the whole paper, and make sure use different symbols for different physical parameters.

Authors response:

Again, we thank the referee for pointing out this omission. In Eq.2 and below we simply removed the redundant definition of the volume of the entire space ?, so now the symbol ? is used only for the Rabi splitting.

3. In “Introduction”, the authors briefly introduced cavity exciton-polaritons with different types of excitonic material, however, organic cavity exciton-polaritons are not mentioned, as this is a very promising topic under much attention, some introduction and citation are needed.

Authors response:

We appreciate this suggestion made by the referee. In the Sec. Introduction we introduced the 2nd paragraph where we cited and briefly explained the recent publications related to the organic microcavity devices.

4. On Page 2, as the equation, ?!" = ??/√?, is first mentioned, what the symbol of Q represent should be provided.

Authors response:

We thank the referee for pointing out this omission. In new version we provided the meaning of the symbol Q after this equation.

5. On Page 2, the authors mentioned “Even though the s(TE) surface or 2D polaritons do exists for some conditions, there is still no experimental evidence of such modes”, however, TE surface 2D polariton has been experimentally demonstrated in both inorganic (quantum well and TMD monolayer) and organic systems with Bloch surface wave (BSW) structures, which are listed as following:
1)
https://aip.scitation.org/doi/10.1063/1.4863853

2) https://www.nature.com/articles/s41565-018-0219-7

3) https://www.nature.com/articles/lsa2016212

4) https://onlinelibrary.wiley.com/doi/full/10.1002/adma.202002127

Authors response:

We thank the referee for pointing out these relevant experimental investigations. Indeed, there is experimental evidence of the hybridization between the s(TE) BSW and excitons in various organic/inorganic addlayers. However, in these experiments there is the hybridization between the photons confined between the semi-infinite dielectric and the Bragg reflector, and the excitons which is a concept slightly different from the one we use here. Our idea is to simulate the strong coupling between a free photon and exciton. For example, the hybridization between the photon and exciton in an organic crystal or TMD in vacuum. The dielectric here serves just as the supporting substrate. We added the relevant references and a short explanation at the end of the 3th paragraph of Sec. Introduction.

6. In Fig. 4(b) and 4(c), only lower polariton states are provides, it will be better to show both upper and lower polaritons, as this is the first time to provide the polariton dispersion results for the anti-crossing.

Authors response:

We thank the referee for this suggestion. We added the upper exciton-polaritons branches in Fig. 4(b) and denoted lower and upper exciton-polaritons branches as LPB and UPB in that figure, as well as in the Figs. 4(c), 5(b) and 5(e), where they were already shown, but not denoted. We also added corresponding text in Sec. 3. Results and discussion.

However, we should emphasize that the anti-crossing behavior and the Rabi splitting to the LPB and UPB is a well-defined concept only when describing the interaction between the well-defined eigen modes, such as excitons and cavity-photons or, as the referee mentioned, BSW. In our case the only well-defined modes (boson) are the excitons ωex1ωex3. On the other hand, the photons appear as a continuum of eigen-modes above the light lines or . This means that here it only makes sense to observe the deformation of the exciton dispersion ωex1 which is characterized by the exciton bending parameter Ω. However, in Figs.5 it seems that the edge of the photon continuum Qc or Qc/\sqrt{\epsilon} behaves just like a well-defined eigen-mode, so we also introduced and marked the LPB and UPB in Fig.4(b) and (c) as they appear Figs.5 (b) and (e) as well.
